# The Role of Extracellular Matrix Expression, ERK1/2 Signaling and Cell Cohesiveness for Cartilage Yield from iPSCs

**DOI:** 10.3390/ijms20174295

**Published:** 2019-09-02

**Authors:** Justyna Buchert, Solvig Diederichs, Ursula Kreuser, Christian Merle, Wiltrud Richter

**Affiliations:** 1Research Center for Experimental Orthopaedics, Heidelberg University Hospital, Schlierbacher Landstr. 200a, 69118 Heidelberg, Germany; 2Center of Orthopaedic and Trauma Surgery, Heidelberg University Hospital, Schlierbacher Landstr. 200a, 69118 Heidelberg, Germany

**Keywords:** induced pluripotent stem cells, cartilage regeneration, mesenchymal stromal cells, extracellular matrix, ERK1/2 signaling

## Abstract

Current therapies involving chondrocytes or mesenchymal stromal cells (MSCs) remain inefficient in restoring cartilage properties upon injury. The induced pluripotent stem-cell (iPSC)-derived mesenchymal progenitor cells (iMPCs) have been put forward as a promising alternative cell source due to their high proliferation and differentiation potential. However, the observed cell loss during in vitro chondrogenesis is currently a bottleneck in establishing articular chondrocyte generation from iPSCs. In a search for candidate mechanisms underlying the low iPSC-derived cartilage tissue yield, global transcriptomes were compared between iMPCs and MSCs and the cell properties were analyzed via a condensation assay. The iMPCs had a more juvenile mesenchymal gene signature than MSCs with less myofibroblast-like characteristics, including significantly lower ECM- and integrin-ligand-related as well as lower α-smooth-muscle-actin expression. This correlated with less substrate and more cell-cell adhesion, impaired aggregate formation and consequently inferior cohesive tissue properties of the iMPC-pellets. Along lower expression of pro-survival ECM molecules, like decorin, collagen VI, lumican and laminin, the iMPC populations had significantly less active ERK1/2 compared to MSCs. Overall, this study proposes that this ECM and integrin-ligand shortage, together with insufficient pro-survival ERK1/2-activity, explains the loss of a non-aggregating iMPC sub-fraction during pellet formation and reduced survival of cells in early pellets. Enhancing ECM production and related signaling in iMPCs may be a promising new means to enrich the instructive microenvironment with pro-survival cues allowing to improve the final cartilage tissue yield from iPSCs.

## 1. Introduction

Due to the lack of vasculature and overall low number of cells, articular cartilage lacks healing capacity upon injury. The cell-based regenerative approaches are considered most promising in restoring properties of native hyaline cartilage. However, current standard treatments, such as microfracture, autologous chondrocyte implantation or their modifications, remain ineffective. Articular chondrocytes, the only cells currently used for clinical cartilage regeneration, are afflicted with an extremely limited availability as well as inherent low proliferation potential. This also critically restricts the application of articular chondrocytes for drug screening and disease modelling. Therefore, new and unlimited sources for articular chondrocytes are urgently needed.

Bone-marrow-derived mesenchymal stromal cells (MSCs) are well known to acquire a chondrogenic phenotype in defined in vitro settings, verified by the detection of collagen type II and proteoglycan deposition [1,2]. Numerous tissue engineering approaches involving MSCs have been investigated in vitro as well as in animal models [3]. However, the cells have only limited ex vivo expansion capacity before their full differentiation capacity is lost [4,5]. Furthermore, MSCs are characterized by undesired intrinsic endochondral commitment manifested by a hypertrophic phenotype during in vitro chondrogenesis. MSC hypertrophy is typically associated with collagen type X and alkaline phosphatase expression and the resulting propensity of the formed tissue to undergo mineralization [6]. Altogether, the biochemical and biomechanical properties of MSC-engineered cartilage remain inferior to the native tissue. Due to these limitations and despite years of research, MSC-derived chondrocytes are no equal source to articular chondrocytes for drug screening and in vitro disease modelling, nor have they been tested in any clinical trial so far (ClinicalTrials.gov as of July 2019) for the treatment of cartilage defects [7,8].

Induced pluripotent stem cells (iPSCs), which can be established from nearly each patient, have been put forward as a promising alternative cell source for articular chondrocytes for such applications. IPSCs are characterized by virtually unlimited proliferation capacity without compromising the differentiation potential, which promises unrestricted tissue yield.

iPSCs are also known to have the capacity to undergo chondrogenic differentiation under defined conditions [9,10,11]. Some strategies for in vitro-based iPSC chondrogenesis require the generation of intermediate mesenchymal progenitor cells (iMPCs) [12,13,14,15]. Compared to MSCs, iMPCs were shown to proliferate faster and to express genes associated with cellular aging at lower levels [12,16]. The successful formation of cartilage-like tissue from iPSCs/iMPCs is evident from the deposition of collagen type II and proteoglycans and some reports suggested that the differentiation with less hypertrophy may be feasible [9,11].

Substantial cell loss during chondrogenesis, is currently a bottleneck in establishing a robust and reliable protocol for articular chondrocytes generated from iPSCs/iMPCs. When subjecting iMPCs to chondrogenic cues, only a minority of cells are left in the end-product cartilage, rendering the method much more ineffective than for MSCs. The reasons for this elevated cell loss remain unclear and this study hypothesized that a thorough comparison of the cell phenotype and activity of expanded iMPCs versus MSCs before the start of chondrogenesis provides functional cues on how to improve chondrocyte generation from iPSCs in the future.

The aim of the study was, therefore, to identify the mechanisms and pathways that could be responsible for the evident high cell loss during iMPC-based in vitro chondrogenesis compared to MSCs. To this end, this study compared the global gene expression profiles of expanded iMPCs and MSCs and captured the most pronounced differences and predictions for pathway activities by bioinformatic-based methods. This study compared the mesenchymal phenotype of iMPCs versus MSCs, excluded iMPC mis-differentiation into other lineages and characterized the most obvious differential cell properties between iMPCs and MSCs by functional assays and measurement of the pathway activity. This knowledge allows the improvement of protocols for efficient chondrocyte generation from iPSCs to provide an unlimited source of chondrocytes for cartilage regeneration, drug screening and disease modeling in the future.

## 2. Results

### 2.1. High Cell Loss Occurs at the Onset of In Vitro Chondrogenesis of iMPCs

When the cartilage was generated from the same number of iMPCs versus MSCs under identical 3D culture conditions, a prominent tissue size difference was evident at the termination of the pellet culture at day 42. Histology revealed a similar staining for collagen type II (Figure 1A) and proteoglycans for both groups (Figure 1B). Histomorphometry of toluidine blue-stained central sections (not shown) demonstrated that iMPC-derived pellets were considerably smaller than MSC-formed pellets. The quantification according to the idealized ellipsoid volume formula using the diameter of the central sections showed that iMPC-derived pellets had a 10.2-fold lower volume than MSC-pellets (Figure 1C).

In line, the DNA content of iMPC-derived pellets dropped to significantly lower levels. While MSC-derived pellets still contained approximately 52% ± 6.5 of the initial DNA amount on day 7, only 14% ± 7.5 of DNA was left in the iMPC-derived pellets (Figure 1D). At day 42, the iMPC-derived cartilage contained only 3% ± 2.4 of the initial DNA, whereas the MSC-pellets maintained 29.4% ± 6.5 of DNA (Figure 1D). The time course experiments during the first week of iMPC chondrogenesis demonstrated a significant cell loss from day 3 on (Figure 1E). Altogether, this demonstrated that iMPCs had a significantly lower ability to contribute to cartilage tissue yield compared to MSCs.

### 2.2. IMPCs Are More Juvenile Mesenchymal Progenitors than MSCs

To search for the reasons for the significantly higher cell loss of iMPCs, global gene expression profiling was performed at the end of the expansion culture using the samples from 4 independently generated iMPC populations and 4 MSC donors. The hierarchical clustering of the complete microarray data set clearly separated the two cell types even without pre-selection for any gene subsets (Figure 2A). The high distance between iMPCs and MSCs demonstrated that the difference between both cell types was substantial, while the individual iMPC populations and MSCs derived from different donors were closely related to each other. The significance analysis of microarrays (SAM) identified 1159 differentially expressed genes (DEGs) between groups (false discovery rate <0.05). Among 534 genes higher expressed in iMPCs compared to MSCs, 99 were elevated more than 3-fold (Appendix A), while among 625 lower expressed genes, 229 were more than 3-fold lower expressed (top 100 shown in Appendix A). Overall, this indicated a net production deficit for many gene products in iMPCs (Figure 2B; Appendix A).

When the differentiation status of iMPCs was examined, the microarray data showed that the expression levels of pluripotency-associated genes characteristic for iPSCs, including *NANOG*, *OCT4* or *ZFP42* were downregulated below the background as expected. Other stem cell markers, such as *SOX2*, *SSEA4*, *SSEA1*, *MYC* and *KLF4* showed expression levels similar to MSCs (not shown). Most endodermal as well as ectodermal markers were below the detection level in iMPCs or similar to MSCs (Appendix A). Thus, no accidental mis-differentiation of some iMPCs into undesired lineages during monolayer expansion was evident. Importantly, iMPCs had comparable expression profiles for a panel of widely known MSC markers, including *CD44*, *CD73* or *CD90*, although cDNA-array-recorded mean gene expression levels of some other mesenchymal markers like *CD105* (*ENG*; −6.9 fold), *CD106* (*VCAM1*; −19.4 fold) and *PDGFRA* (−10.5 fold) remained significantly lower (Table 1).

The flow cytometry indeed revealed hardly any iMPCs positive for PDGFRA and CD106 (VCAM1), while CD105 was expressed on approximately half of the cells (46.7% ± 12.7) compared to 95.6% ± 5.7 in MSCs (Figure 3A,B).

Interestingly, CD105 and CD106 were lower expressed in fetal and perinatal MSCs compared to the adult cells [17,18,19,20], indicating that iMPCs adopted a more juvenile phenotype than adult MSCs. In line, a typical marker for early mesoderm *MSX1* [21] was significantly elevated in iMPCs (5.9 fold) and a set of previously defined aging markers [22] was significantly higher in MSCs, including *TMEM119* (−15.2 fold), *ENPP2* (−15.0 fold), *COX7A1* (−12.6 fold) (Appendix A) and *EYA2* (−2.6 fold, data not shown). Overall, these results demonstrated a successful mesenchymal differentiation and no mis-differentiation of iPSCs during iMPC generation and indicated that iMPCs are more juvenile mesenchymal progenitors than MSCs.

### 2.3. Distinct ECM Expression Profiles of iMPCs and MSCs

To better classify the main differences in gene expression between iMPCs and MSCs, the microarray data were subjected to PANTHER analysis [23]. When all significantly differentially expressed genes (DEGs) were classified according to the Panther protein class category, the ECM structural protein class (PC00103) as well as the ECM protein class (PC00102) were significantly overrepresented (up to 4.9 fold, *p* < 0.001; Figure 4A).

Interestingly, out of the 18 DEGs from both classes, 16 were lower expressed in iMPCs (Table 2). The ECM-related genes were also most significantly enriched in the gene enrichment tests for the Gene ontology cellular component and the Gene ontology biological process annotation data sets or on the second place when tested for the Gene ontology molecular function (not shown). Altogether, this suggested lower ECM production activity of iMPCs compared to MSCs in the expansion culture.

### 2.4. Differential Expression of Integrin Signaling Network Members

Next, the gene enrichment test according to the Pathway category was performed with all significant DEGs using the fold-difference in expression levels as an input. The genes belonging to the Integrin signaling pathway (P00034) were most significantly enriched in this analysis (Table 3).

Again, most of the genes of this category were significantly lower expressed in iMPCs (14 out of 22; Appendix A) with *COL6A1/2/3* (−8.4, −5.2 and −10.5 fold respectively), *COL16A1* (−6.0 fold), *COL8A1* (−4.2 fold) and *ITGB5* (−2.4 fold) being prominent examples. Interestingly, α-smooth muscle actin (*ACTA2*), a typical marker for the activated fibroblasts was also significantly lower expressed in iMPCs (−10.4 fold), indicating a less myofibroblastic phenotype of iMPCs. In contrast, *COL4A5* (7.0 fold) and *COL13A1* (5.3 fold) were higher expressed in iMPCs. The position of these genes on a volcano plot was visualized in Figure 4B which also pictured a considerable number of prominent ECM-related genes not annotated by the Panther tool like *DCN* (−23.9 fold), *LAMA4* (−14.1 fold) and *LUM* (−10.6 fold). When the differential expression of selected genes was verified by qPCR in partly-independent samples (Figure 4C), even higher differences were obvious between iMPCs and MSCs. Taken together, the most differential cell properties between the more juvenile iMPCs and adult MSCs were a reduced ECM-related gene expression, lower activated fibroblast markers and an altered integrin signaling-related network, which may result in differential cell-cell and cell-matrix adhesive properties of iMPCs versus MSCs.

### 2.5. Distinct Aggregation Behavior of iMPCs versus MSCs

To test whether the cell-cell aggregation, substrate adhesion and spontaneous chondrogenic pellet formation would be different in iMPCs versus MSCs, both cell types were seeded at high-density on non-coated adhesive culture plastic in a chondrogenic medium. When the cell aggregation behavior was recorded in regular intervals over 7 days, iMPCs rapidly formed multiple free-floating cell-cell aggregates with only low cell numbers adhering to plastic. In contrast, MSCs rapidly formed a plastic-adhered cell multilayer with no floating cell-cell aggregates. The contraction of the cohesive MSC multilayer sheet into one large non-adherent chondrogenic pellet took 5 to 9 days (mean of 6.5 days; Figure 5A).

The iMPCs completed their aggregation into several smaller cell aggregates (3 out of 5 independent experiments; Figure 5B) or one large aggregate per well (2 out of 5 experiments, not shown) within 2–6 days (mean 3.2 days) without going through a plastic-adherent phase. Thus, most iMPCs preferred the cell-cell aggregation above the surface adherence. This was possibly as reminiscence of the colony growth of iPSCs which do not tolerate the monolayer status.

To possibly link the ECM/integrin gene expression to the efficient aggregate formation of the cells, pellet-forming and non-aggregating iMPCs were carefully separated by aggregate picking on day 2 and mRNA levels of *COL6A3*, *DCN* and *ITGA2* were compared. The pilot data suggested that aggregate-forming cells expressed 3.5-fold more *COL6A3*, 2.6-fold more *DCN* and 4.1-fold more *ITGA2* than the non-aggregating cells (*n* = 1). Overall, the enhanced ECM and integrin expression in MSCs correlated with the improved cell aggregate formation. This may explain the differential cell aggregation properties of iMPC subpopulations. The loss of non-aggregating iMPCs due to the insufficient ECM/integrin gene expression may, thus, be one important reason for the reduced cell content of iMPCs versus MSC pellets.

### 2.6. Lower ERK1/2 Signaling in iMPCs versus MSCs

To illuminate the consequences of lower ECM- and integrin-related gene expression, as well as inferior cohesivity of iMPCs on the signaling pathway activity in iMPCs versus MSCs, the main integrin-signaling and adhesion-relevant pathway, extracellular signal-regulated kinase 1/2 (ERK1/2/MAPK) [24] was analyzed by western blotting in iMPCs versus MSCs. Interestingly, iMPCs showed no phosphorylated pERK1/2 or levels close to the detection limit, while MSC lysates of the same protein content showed strong bands on the same blot with significantly increased pERK1/2 to β-actin levels (Figure 6A).

To assess a possible functional link between better cohesion and activation of pERK1/2, the ability of MSCs to adhere to the substrate or to one another by transferring them into 3D agarose culture was reduced. In agarose, the cells can no longer interact as in 2D culture and remain round as typical for suspended cells, which were used as a control. Indeed, the low pERK1/2 levels observed in suspension (c) were maintained after a shift to the agarose culture, while pERK1/2 levels increased significantly when the cells were allowed to adhere to plastic in 2D under otherwise identical conditions. Overall, pERK1/2 levels were 9.8-fold lower (*p* < 0.001) in agarose than in the parallel monolayer cultures (Figure 6B). This supports a positive correlation between the adhesion and pERK1/2 activity as an established pro-survival signal for many cells. Taken together, this suggested that due to the lower cell cohesion ability of iMPCs, pro-survival pERK1/2 signaling remained low, rendering the cells more vulnerable to stress-induced cell death after the shift to chondrogenic culture conditions.

## 3. Discussion

The iPSCs with unrestricted proliferative and chondrogenic differentiation potential are an attractive source for the generation of human articular chondrocytes, which are urgently needed for cartilage tissue regeneration, drug screening and disease modelling strategies. One bottleneck of iPSC chondrogenesis is, however, the significant cell loss of the intermediate iPSC-derived mesenchymal progenitors during chondrogenic culture conditions. In a search for potential reasons for this vulnerability, this study identified a higher early mesodermal marker and a lower aging marker expression in iMPCs. This suggested a more juvenile mesenchymal progenitor cell phenotype in iMPCs versus adult MSCs, which was further characterized by significantly lower levels of many ECM-related genes. The integrin-signaling pathway-related molecules were discovered as the most distinctive mark between the expanded iMPC and MSC populations and in line, it was observed that iMPCs preferred the cell-cell aggregation in a high cell density aggregation assay under chondrogenic conditions, while MSCs preferred substrate adhesion and multilayer cell sheet formation. The observed lower cohesiveness of iMPCs obviously led to a non-aggregating sub-fraction of the cells which did not contribute to chondrogenic pellet formation and expressed much lower levels of ECM-related *DCN*, *COL6A3* and *ITGA2*. The loss of these non-aggregating iMPCs was an important reason for substantial cell decline within the first days. In correlation with the lower expression levels of ECM-related genes and integrin pathway members, pERK1/2 activity was significantly lower in the expanded iMPC versus MSCs. The iMPCs, thus, lacked an important pro-survival stimulus to tolerate the shift to low nutrient, hypoxic, high density culture conditions, which are typical for chondrogenic culture. This study concludes that, due to a less myofibroblastic phenotype of iMPCs and an ECM-poor microenvironment, a fraction of iMPC cannot contribute to cartilage pellet formation due to the insufficient cell-cell and cell-ECM cohesion properties while others may die due to insufficient pro-survival pERK1/2-signaling. Guiding iPSCs towards mesenchymal progenitors with enhanced ECM production, adhesion properties and pro-survival pERK1/2 signals before a shift to chondrogenic conditions is a new goal to prevent massive cell loss during early iMPC chondrogenesis and improve articular chondrocyte neogenesis from human iPSCs.

During our search for potential reasons for the high cell loss during iPSC chondrogenesis, this study excluded that a substantial residual iPSC fraction existed among iMPCs, which may die during early chondrogenesis. It was also verified that the cell loss did not result from pronounced mis-differentation of iPSCs into endodermal or ectodermal cells, which may then no longer be able to develop into chondrocytes. The common mesenchymal marker profile of iMPCs was comparable to MSCs. This is in line with existing studies characterizing iMPCs [14,15]. The low CD106 expression in iMPCs could be related to the tissue of origin of iPSCs, as it was shown in previous work [15]. Although PDGFRα was used in several studies as a selection marker for the enrichment of MSCs or iMPCs with higher chondrogenic differentiation potential [25,26,27], a lack of PDGFRα surface expression does not prevent the successful chondrogenic differentiation, in accordance with other groups [11,28,29]. Interestingly, the differential expression of several mesenchymal genes in iMPCs matched the phenotypes of fetal or adult MSCs. This in line with the literature reporting the rejuvenation of iMPCs with regard to age-related DNA methylation [16,22,30].

By means of global transcriptome analysis, it was found that the expanded iMPCs and MSCs have highly distinct ECM- and integrin network gene expression profiles. Integrin-related signaling is important during spheroid formation and the initial steps of cellular condensation during in vitro chondrogenesis [31,32,33,34,35], and therefore is highly relevant in the context of iMPC aggregation. However, the integrin pathway-related DEGs contained only a few integrin subunits. This suggests that the integrin receptor profile itself is rather similar in iMPCs and MSCs. However, integrin activation is regulated via numerous other factors than transcriptional regulation, including the dimerization status, heterodimer type, conformational changes, clustering followed by focal adhesion formation or the presence of cations and adaptor proteins [36]. Therefore, the recruitment and activation of integrin downstream effectors, such as focal adhesion kinase (FAK) or paxillin, is required to finally conclude about the similarities in integrin signaling. Importantly, the integrin network gene list enclosed a panel of prominent ECM genes that were consistently lower expressed in iMPCs than in MSCs and suggested that iMPCs may encounter a shortage of ECM ligands for integrin receptors at start of chondrogenesis. To the authors’ knowledge, this is the first study indicating that ECM substantially differs in composition between iMPCs and MSCs.

ECM provides structural support, anchorage, serves as a reservoir for growth factors, directs differentiation as well as maintains cell survival [37,38]. Remarkably, all 3 chains (A1, A2 and A3) of collagen type VI were significantly lower expressed in iMPCs. Collagen VI regulates mechanotransduction and tissue biomechanical properties [39,40] and can confer pro-proliferative and pro-survival signals via the AKT or PI3K signaling pathways [41]. Decorin and lumican bind to collagen, acting as linkers and facilitate correct organization of fibrils, enabling the formation of functional collagen networks [42,43,44,45]. Moreover, both proteoglycans were demonstrated to enhance cell survival, decorin at the early stages of matrix assembly [46] and lumican in the context of cardiac fibroblasts, where it also reinforced the myofibroblastic phenotype [44]. In turn, laminin α4 chain was demonstrated to be involved in cell cluster formation [47]. Overall, this suggests iMPCs may not only experience less ECM with an altered structure, but the insufficient integrity of an ECM may provide less pro-survival support at the stressful shift to low oxygen and low nutrient high density culture.

In line with this, an important finding of this study was the significantly lower pERK1/2 level in iMPCs compared to MSCs upon expansion. Of note, pERK1/2 levels varied between individual MSC donor populations. This is expected when primary MSCs from different donors are compared to iMPCs derived from one iPSC line. To avoid this limitation, the testing of iMPCs from different iPSC lines should be performed in the future. Notably, ERK1/2 is a known downstream target of adhesion, and in particular, integrin-mediated signaling. In fact, integrin-mediated organization of actin cytoskeleton supports the translocation of ERK1/2 into the nucleus, coupling substrate anchorage with ERK1/2 stimulation [48]. Our data demonstrated that high pERK1/2 levels in MSCs are indeed a consequence of substrate adhesion, since MSCs cultured in the suspension or inert agarose abrogated ERK1/2 phosphorylation. The ERK1/2 signaling has a major role in providing the pro-survival signals, both by the inactivation of pro-apoptotic proteins, such as FOXO3a, as well as by the activation of RSK kinases, that further activate an array of pro-survival factors, such as c-Fos, c-Jun, Bad, IκB or C/EBPβ [49]. Therefore, insufficient anchorage-dependent ERK1/2 signaling in iMPCs may be deleterious and make cells susceptible to the shift in culture conditions. Importantly, low ERK1/2 signals can affect cell survival both during cell aggregation as well as upon completion of aggregate formation.

The implemented condensation assay provided an unsophisticated method to compare cell behavior during aggregate formation and, in principle, reflects cell aggregation capacity in the in vitro chondrogenesis assay. The spontaneous cell self-assembly, characteristic for MSCs, is governed by complex cell-cell and cell-matrix interactions [50], as well as by the active role of ECM [50,51]. Furthermore, cytoskeleton-transduced adhesion-related signaling governs cell survival, proving differential ECM and integrin network expression profiles as well as lower α-smooth muscle actin (ACTA2) expression levels in iMPCs. This is highly relevant for the aggregation processes. Indeed, ACTA2 is not only a major myofibroblastic marker, but it is also an important downstream integrin effector. It was previously shown that ACTA2 facilitated integrin signaling and efficient cell contraction, leading to ERK1/2 phosphorylation in myofibroblasts [52]. Consistently, the contraction of ACTA2-expressing MSCs facilitated aggregation and therefore enabled the efficient pellet formation [53]. Future research is expected to also include more refined testing of the cell-cell adhesive properties at a single cell level using atomic force microscopy [54] or by the quantification of cell cohesion strength using tissue surface tensiometry [55,56]. These sensitive measurements strengthen the authors view on the differential cell type-dependent cohesiveness and allow better evaluation of a given intervention to enhance the iMPC aggregation process. Taken together, iMPCs had less myofibroblastic traits than MSCs, characterized by scarce ACTA2 and ECM- or integrin network-related gene expression, which correlated with the failure to form cohesive condensing cell sheets.

Notably, the cell aggregation during in vitro chondrogenesis is essentially a spontaneous selection process for the cells with higher potential to form the cartilage tissue end-product. This is reminiscent of previously reported laborious selection methods for iMPCs, such as the enrichment of pre-differentiated cells via embryoid bodies [13,57,58,59,60,61], prospective selection of chondroprogenitors based on a marker expression [25,62] or chondroprogenitor enrichment via genetic engineering [9,11]. Overall, the major goal is to reduce heterogeneity within the starting cell population by directing cells into the beneficial phenotype.

There are several approaches that could increase the cell fraction characterized by higher ECM- and integrin-related gene expression as well as more active ERK1/2 signaling. Although using gel-like biomaterials proved effective for human MSCs [63], the method was not suitable for iMPCs (unpublished data). This is probably due to the lack of adhesion-dependent ERK1/2 activation. In contrast, the stimulation of native ECM synthesis with TGFβ during expansion or using integrin-binding motif (RGD)-enriched exogenous matrices for pellet formation could potentially enhance pro-survival ERK1/2 activity and diminish cell loss [64,65]. Applying decellularized matrices as chondroinductive scaffolds emerges as a novel approach in the efforts to improve chondrogenesis [66,67,68,69]. As surface coatings, such materials could enhance adhesion-dependent pro-survival signaling in iMPCs during expansion or support early pellet formation when used as scaffolds. Alternatively, ERK1/2 signaling could be activated directly in expanded cells via specific molecules. There are several ERK1/2 agonists, such as phorbol 12-myristate 13-acetate (PMA), anisomycin, monocyte chemoattractant protein-1 (MCP-1) or 12-O-Tetradecanoyl-phorbol-13-acetate (TPA) [70,71,72,73,74]. Otherwise, the inhibitors of phosphatases, including orthovanadate or pervandate, or agonists of cAMP, such as cilostamide can be used to boost ERK1/2 activity [75]. This study hypothesized that enhancing ERK1/2 signaling in the iMPCs starting population using a carefully chosen dose and appropriate timing of a given modulator, could improve cell survival in the initial phase of in vitro chondrogenesis.

Taken together, it was proposed that the initial cell loss during iMPC in vitro chondrogenesis may have two reasons. One is the impaired cell aggregation due to lower ECM-, integrin network- and myofibroblastic-related gene expression during pellet formation together with a poor cell contraction resulting in an inferior cohesion of the forming aggregates. A second underlying mechanism could be that during, as well as upon the completion of the aggregation process, iMPC survival may be compromised by insufficient pro-survival cues from the ECM and by the significantly lower levels of anchorage-dependent pro-survival ERK1/2 activity. Our findings underscore the importance of ECM as an instructive microenvironment that supports cell survival. The measures to improve chondrocyte yield from iPSC may include the stimulation of ECM production of iMPCs, the use of exogenous matrices or the direct activation of ERK1/2 signaling in an effort to increase iMPCs cohesion properties and cell survival to allow more efficient chondrocyte generation from iPSCs for regenerative and research applications.

## 4. Materials and Methods

### 4.1. MSC Isolation and Expansion

The MSCs were isolated from fresh bone marrow aspirates obtained from patients undergoing total hip replacement or osteotomy, as previously described [2]. Written consent was obtained from all tissue donors and the study received approval from the local ethics committee (Medical Faculty of Heidelberg S-499/2014 from 17 November 2014). The mononuclear cell fraction was separated from bone marrow aspirates by Ficoll-Paque^™^ density gradient and seeded into 0.1% gelatin-coated culture flasks in an MSC expansion medium (DMEM high glucose w/o L-glutamine, 12.5% fetal bovine serum, 2 mM L-glutamine, 1% non-essential amino acids, 0.1% β-mercaptoethanol [all from Gibco, Invitrogen Life Technologies, Karlsruhe, Germany], 100 units/mL penicillin, and 100 mg/mL streptomycin [Merck-Millipore, Darmstadt, Germany], supplemented with 4 ng/mL basic fibroblast growth factor [bFGF, Active Bioscience, Hamburg, Germany]). After 24 h at 37 °C, 6% CO_2_ in a humidified atmosphere, the nonadherent cells were removed by washing with PBS. At 80% confluence, the cells were detached with 0.05% Trypsin / 0.02% EDTA and re-seeded at the density 5 × 10^3^ cells/cm^2^. The medium was replaced thrice a week up to passage 3 and the cells were characterized according to MSC criteria [76]. 

### 4.2. IPSC Culture

The iPS(IMR90)-4 cell line (WiCell, Madison, USA), generated from foreskin fibroblasts of a healthy donor [77], was routinely cultured on Matrigel^®^ hESC-qualified matrix (Corning Life Sciences, Berlin, Germany) with mTeSR™-1 medium (Stemcell Technologies, Cologne, Germany). The medium was exchanged every day. The confluent iPSCs were detached using 1 mL dispase (1 unit/mL, Stemcell Technologies) and reseeded at a 1:6 to 1:10 ratio. The generation of iMPCs was carried out as described before [10]. In brief, the medium of 50–70% confluent iPSCs colonies was aspirated and the MSC expansion medium was added (DMEM high glucose w/o L-glutamine, 12.5% fetal bovine serum, 2 mM L-glutamine, 1% non-essential amino acids, 0.1% β-mercaptoethanol [all from Gibco, Invitrogen], 100 units/mL penicillin, and 100 mg/mL streptomycin [Merck-Millipore], supplemented with 4 ng/mL bFGF). The medium was exchanged every day. On day 7, the cells were detached with 0.05% Trypsin/0.02% EDTA (Merck-Millipore) and reseeded into 0.1% gelatin-treated culture flasks at a density of 20,000 cells/cm^2^ with MSC expansion medium supplemented with 10 µM ROCK inhibitor Y27632 (Miltenyi Biotec GmbH, Bergisch Gladbach, Germany) for 24–48 h as passage 0. The medium was exchanged with MSC expansion medium (no ROCK inhibitor) thrice a week. Upon confluency, the cells were trypsinized and subcultured for 3 passages.

### 4.3. Transcriptome Analysis

The confluent cells were harvested at passage 3 (*n* = 4). Per sample 5 × 10^5^ cells were subjected to total RNA isolation as described below. The quality control of total RNA, labeling, array hybridization, and microarray scanning were performed at the German Cancer Research Center Genomics Core Facility (Heidelberg, Germany). Gene expression profiling was performed using BeadChip HumanHT-12 v4 as described previously [78].

### 4.4. Bioinformatic Analysis

The fluorescence values from the cDNA array analysis were quantile normalized, log2 transformed and analyzed in Multiexperiment Viewer (MeV) 4.9.0 (TM4 Microarray-Software-Suite) [79]. The genes differentially expressed between iMPCs and MSCs were identified by unpaired Significance Analysis of Microarrays (SAM Version 1.0) [80]. The median false discovery rate (FDR) was set at <0.05 and delta set to 5.336. The gene overrepresentation test according to the protein class category as well as the gene enrichment test were performed with the PANTHER software (http://pantherdb.org, Version 14.1, released 12 March 2019; [23,81], using the list of all significantly differentially expressed genes identified by SAM, excluding 10 histocompatibility (HLA) complex genes (1149 genes in total).

### 4.5. Chondrogenic Induction

The cells were harvested at passage 3 and subjected to 3D high density culture with 5 × 10^5^ cells per pellet. The pellets were cultivated in a chondrogenic induction medium consisting of high-glucose DMEM supplemented with 0.1 µM dexamethasone, 0.17 mM ascorbic acid-2 phosphate, 5 µg/mL insulin, 5 µg/mL transferrin, 5 ng/mL selenous acid, 1 mM sodium pyruvate, 0.35 mM proline, 1.25 mg/mL bovine serum albumin, 1% penicillin/streptomycin and 10 ng/mL recombinant human transforming growth factor 1 (TGF-β1, Peprotech, Hamburg, Germany) for up to 42 days. The medium was changed three times a week.

### 4.6. Aggregation Assay

The 5 × 10^5^ cells in passage 3 were seeded in triplicate onto an uncoated 48-well plate in a chondrogenic induction medium. The spontaneous aggregation of the cells was recorded after 6 h and then every 24 h until aggregation was accomplished. Three representative photos per well were taken and aggregation was considered completed from the time point when no more changes were observed.

### 4.7. 3D agarose Culture

The confluent MSCs in passage 2 were re-suspended in the expansion medium and mixed with 3% agarose solution previously pre-warmed to 39 °C. The agarose-cell suspension was cast into silicon moulds (5 × 10^5^ cells per 25 µL per construct). The constructs were incubated for 3 h at 37 °C with an expansion medium in a humidified chamber with 6% CO_2_ and harvested by immediate submersion in liquid nitrogen and lysed using lysis buffer. The control cells were seeded onto a gelatin-coated tissue culture plastic in monolayer at a density of 5 × 10^5^ cells/cm^2^, incubated for 3 h at 37 °C and harvested in ice-cold lysis buffer.

### 4.8. Quantitative Gene Expression Analysis

Total RNA was isolated by a standard guanidinium thiocyanate/phenol extraction procedure using peqGOLD TriFast^™^ (Peqlab, Erlangen, Germany) according to the manufacturer’s protocol. The synthesis of cDNA was performed with 200 ng of total RNA as a template using Omniscript reverse transcriptase (0.2 U/µL), oligo(dT) primer (1 µM, both Qiagen, Hilden, Germany) and a ribonuclease inhibitor (RNaseOUT, 40 U/µL, Invitrogen Life Technologies) according to the manufacturer’s instructions. The relative gene expression levels were determined by quantitative polymerase chain reaction (qPCR) analysis using SybrGreen (ThermoFischer Scientific, Schwerte, Germany) and the LightCycle^®^96 (Roche Diagnostics, Rotkreuz, Switzerland) as well as the gene-specific primers shown in Appendix A. The specificity of the PCR products was confirmed by a melting curve analysis and agarose gel electrophoresis of PCR products. The gene expression was normalized to the mean C_t_ values of the reference genes CPSF6, HPRT and RPL13. The relative difference in expression levels was calculated as 1.8^-ΔCt^.

### 4.9. Quantification of DNA Content

The DNA content of the cells was determined using the Quanti-iT PicoGreen dsDNA kit (Invitrogen Life Technologies) according to the manufacturer’s instructions. The cells were pre-digested in a lysis buffer (Phosphate Buffer Saline, 0.1% Triton-X 100) over night at 60 °C, sonificated and analyzed by mixing 20 µL of the digested sample with 80 µL TE buffer (200 mM Tris-HCl, 20 mM EDTA) and 100 µL PicoGreen solution. The standards were prepared from λ-DNA and DNA content was determined by fluorescence measurement at 485/535 nm.

### 4.10. Flow Cytometry

The 2 × 10^5^ cells were fixed and labeled with anti-CD105 (1:40; Miltenyi Biotec) or anti-CD106 (1:25; Biorad, Dreieich, Germany) or anti-CD73 (1:40; BD Pharmingen, Heidelberg, Germany) or anti-PDGFRα (1:25; BD Pharmingen) PE-conjugated antibodies. The surface marker positive cells were quantified in a MACSQuant Flow Cytometer and the results were analyzed using MACSQuantify Version 2.11.1746.19438 (both Miltenyi Biotec).

### 4.11. Histology

The pellets were fixed in 4% formaldehyde for 2 h, dehydrated and paraffin-embedded. The sections (5 µm) were deparaffinized, rehydrated and stained with Safranin O (0.2% in 1% acetic acid) and fast green (0.04% in 0.2% acetic acid) or with 0.1% toluidine blue solution as described previously [82,83]. Immunohistological staining was performed as described previously [78]. Briefly, the sections were pretreated with 2 mg/mL hyaluronidase (Merck-Millipore) and 1 mg/mL pronase (Roche Diagnostics). PBS containing 5% bovine serum albumin was used to block non-specific background. The sections were incubated overnight at 4 °C with a 1:1000 diluted monoclonal mouse anti-human collagen type II antibody (II-4C11, ICN Biomedicals, Aurora, Ohio, USA) in PBS containing 1% BSA. The reactivity was detected using biotinylated goat anti-mouse secondary antibody (1:500; 30 min, at room temperature; Dianova, Hamburg, Germany), streptavidin-alkaline phosphatase (30 min, at 20 °C, Dako, Hamburg, Germany) and fast red (Sigma-Aldrich, Taufkirchen, Germany).

### 4.12. Histomorphometry

The pellet volume was measured using photographs of the histological pellet sections stained with toluidine blue and analyzed with ImageJ Software (National Institutes of Health, Bethesda, MD, USA) [84]. The scale was set according to the scale bar at the photograph and the diameter was measured using the standard ImageJ measure function. The approximate volume was calculated according to the sphere volume formula V = (4/3) × π × r^3^, where the radius was half of the mean of the horizontal and vertical pellet section diameters.

### 4.13. Western Blotting

The cells were harvested on ice in PhosphoSafe^™^ Extraction Reagent and 1 mM Pefabloc^®^SC (both Sigma-Aldrich) and lysates were centrifuged at 13,000 rpm and 4 °C for 20 min. The concentration was determined using Bradford assay kit (Sigma-Aldrich). The lysates were then separated by SDS-PAGE gel electrophoresis and transferred to a nitrocellulose membrane (GE Healthcare; Munich, Germany). After blocking for 1h at RT with 5% skim milk in 0.1% TBST (25 mM Tris/HCl pH 7.4; 145 mM NaCl, 2.7 mM KCl, 0.1% Tween 20), the membrane was incubated over night at 4°C with a primary antibody. The β-Actin (1:10000, GeneTex, GTX26276/18985), pERK1/2 (1:200, SantaCruz sc/7383/G0518) and ERK1/2 (1:1000, Cell Signaling 9102/26) antibodies were diluted in 5% skim milk in 0.1% TBST. Next, the blots were washed with 1% TBST and incubated with anti-mouse (Jackson Immuno Research, 115-035-071/ 29454) or anti-rabbit (Jackson Immuno Research, 111-035-046/27362) secondary antibodies in 5% skim milk in 0.1% TBST for an hour at RT. After washing with 0.1% TBST, the bands were visualized with ECL (Roche Diagnostics).

### 4.14. Statistical Analyses

For the microarray analysis, see Section 4.4 Bioinformatics analysis. For all other data, the mean and standard deviations were calculated. The differences between the two groups were assessed with Mann-Whitney U-test. The regulation over time within one group was assessed with the Kruskal-Wallis test and post-hoc Mann-Whitney U-tests.

## Figures and Tables

**Figure 1 ijms-20-04295-f001:**
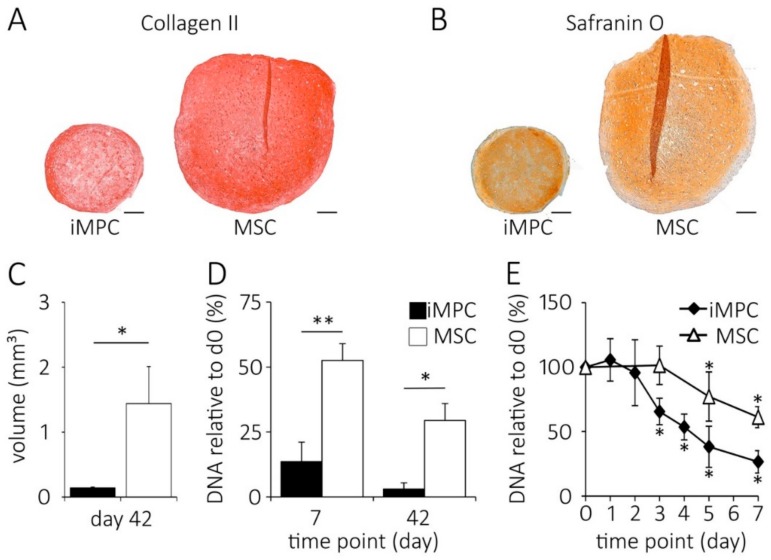
Size and DNA content of intermediate mesenchymal progenitor cells (iMPC)- versus mesenchymal stromal cells (MSC)-derived cartilage pellets. The 5 × 10^5^ cells were subjected to chondrogenesis for the indicated time points and processed for histology and DNA quantification. (**A**) Collagen type II immunostaining and (**B**) Safranin O staining (representative staining of iMPC and MSC pellets, *n* = 4 independent donor populations per group, scale bar = 200 µm). (**C**) Tissue volume at day 42 calculated from histomorphometric data of iMPC- and MSC-derived cartilage pellets (*n* = 6 donor populations per group; iMPC [black bars], MSC [white bars] mean ± standard deviation; * *p* < 0.05 between groups, Mann-Whitney U-test). (**D**) The relative DNA content of pellets with day 0 set as 100% (*n* = 4-13 samples per group; mean ±standard deviation; * *p* < 0.05, ** *p* < 0.01 between groups, Mann-Whitney U-test). (**E**) Time course of DNA loss within the first week of chondrogenesis (*n* = 3 independent iMPC or MSC populations; * *p* < 0.05, compared to day 0, Kruskal-Wallis with post-hoc Mann-Whitney U-tests; the mean values ± standard deviation).

**Figure 2 ijms-20-04295-f002:**
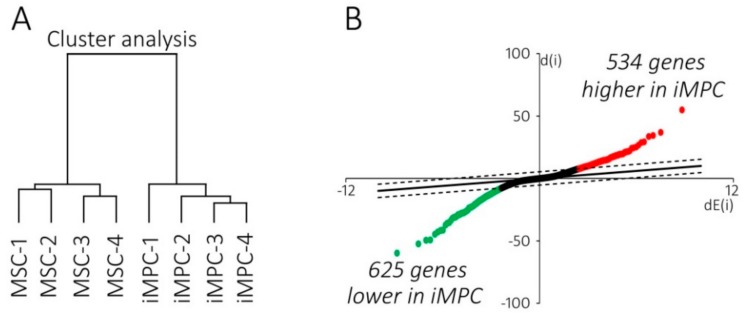
The gene expression profiling in iMPCs versus MSCs. The total RNA extracted at the end of passage 3 from 4 independent iMPC and MSC populations were subjected to genome-wide cDNA microarray analysis. (**A**) Cluster analysis of the sample set based on whole-genome expression data. (**B**) Significance analysis of microarrays (SAM) of global expression data depicted as scatter plot. The observed relative difference d(i) was plotted against the expected relative difference dE(i). The dashed lines define the difference between d(i) and dE(i) beyond which genes are considered significant. The red and green points denote genes significantly higher or lower expressed in iMPC compared to MSC, respectively.

**Figure 3 ijms-20-04295-f003:**
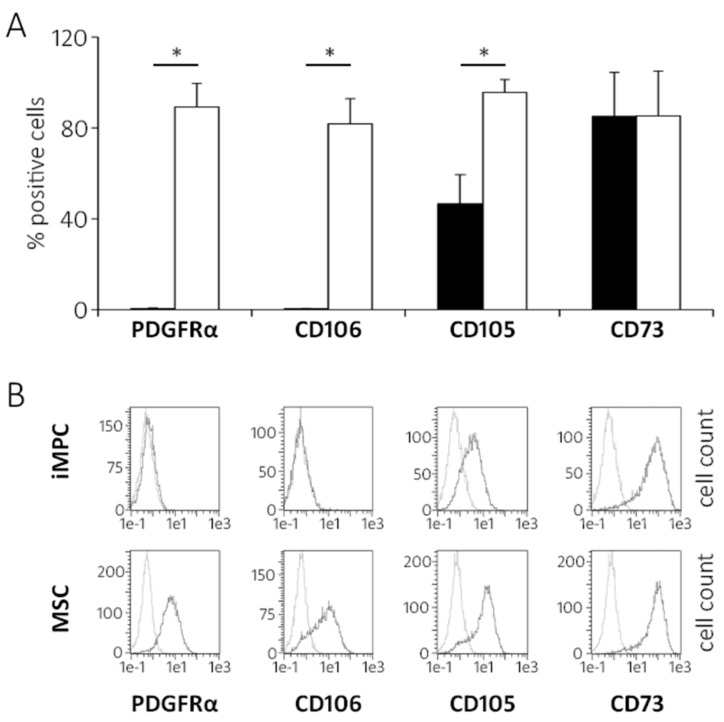
The surface expression of selected mesenchymal markers in iMPCs versus MSCs. Passage 3 cells were harvested, fixed, stained with antibodies and subjected to flow cytometry. (**A**) The proportion of mesenchymal marker-positive cells of iMPCs and MSCs (*n* = 3 independent iMPCs [black bars] and MSC [white bars] donor populations; the mean ± standard deviation; * *p* ≤ 0.05 between groups, Mann-Whitney U-test). (**B**) The representative histograms; light grey lines represent unstained control cells. The gates were adjusted to the control cell population to determine the percentage of positively stained cells.

**Figure 4 ijms-20-04295-f004:**
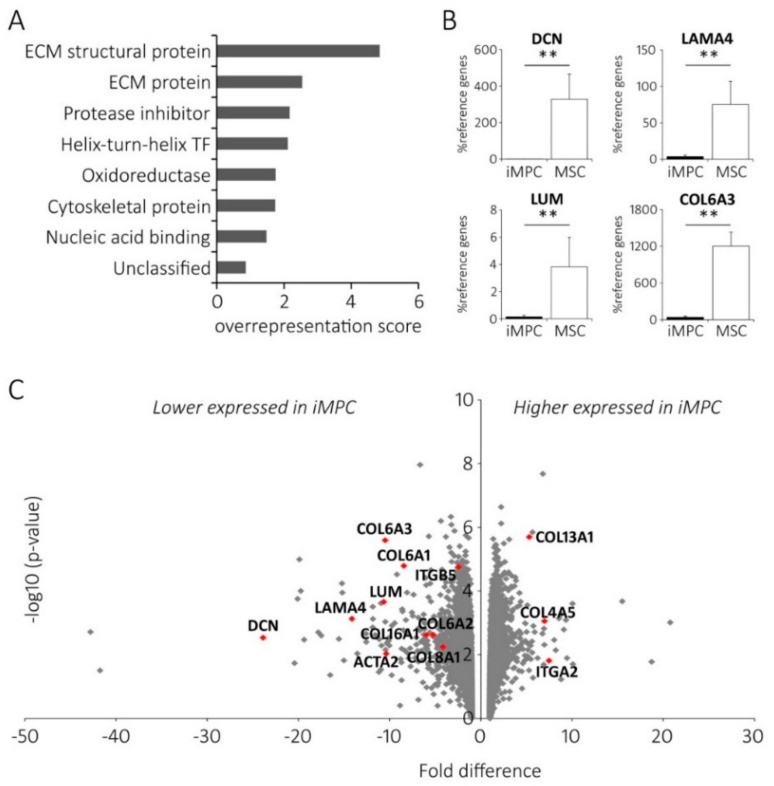
Differential gene expression analysis between iMPCs and MSCs. (**A**) Genome profiling data were subjected to a Panther analysis, displaying statistical overrepresentation of the DEGs classified according to the protein class category. The binomial test and false discovery rate were used for statistics. (**B**) RNA from confluent passage 3 cells was extracted and cDNA was subjected to a qPCR analysis of selected ECM genes with the highest fold expression differences. *RPL13*, *HPRT* and *CPSF6* served as housekeeping reference genes (*n* = 6 independent iMPC [black bars] and MSC [white bars] donor populations; the mean ± standard deviation; ** *p* < 0.01 between groups, Mann-Whitney U-test). (**C**) The volcano plot depicting the fold differences in gene expression plotted against the negative logarithm of the statistical significance of gene expression differences (Student’s T-test) of all genes. The genes mentioned in the text are highlighted.

**Figure 5 ijms-20-04295-f005:**
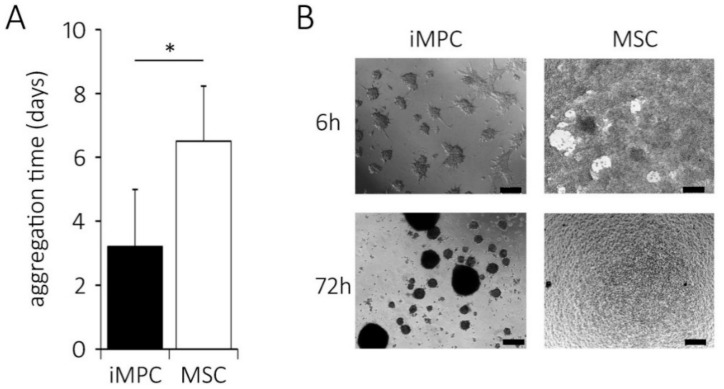
The condensation assay for iMPCs and MSCs. The 5 × 10^5^ cells were seeded onto the uncoated tissue culture plastic in a chondrogenic medium supplemented with 10 ng/mL of TGFβ. (**A**) The completion of aggregate formation was judged visually in standardized time intervals. (*n* = 5 independent iMPC populations [black bar] and *n* = 4 independent MSC donor populations [white bar]; mean ±standard deviation; * *p* < 0.05 between groups, Mann-Whitney U-test). (**B**) The aggregate morphology at 6 h and 72 h after seeding (scale bar = 200 µm).

**Figure 6 ijms-20-04295-f006:**
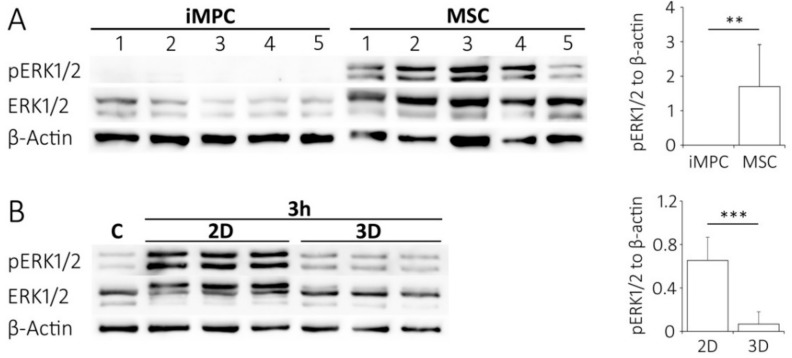
Western blot analysis of pERK1/2 levels in iMPCs and MSCs. (**A**) The expanded cells were harvested at passage 3 and pERK1/2 levels were analyzed by Western blotting using β-actin as loading control (*n* = 5 independent iMPC populations 1–5 and MSC donor populations 1–5). Band intensities were quantified by densitometry. (**B**) The expanded cells were harvested at passage 2 and analyzed for pERK1/2 expression in cell suspension as a control (c) or 3 h after shift to monolayer (2D) or agarose culture (3D). The band intensities were quantified by densitometry (*n* = 3 MSC donor populations with 3 technical replicate cultures each). (Mean ± standard deviation; ** *p* < 0.01, *** *p* < 0.001 between groups, Mann-Whitney U-test).

**Table 1 ijms-20-04295-t001:** Mean microarray expression levels of typical mesenchymal markers in expanded iMPCs and MSCs.

Gene Symbol	iMPC	MSC	Fold Difference	Gene Symbol	iMPC	MSC	Fold Difference
*CD10 (MME)*	160	b	1.7	*CD166 (ALCAM)*	2174	937	2.3
*CD14*	87	171	2.0	*CD340 (ERBB2)*	292	237	1.2
*CD29 (ITGB1)*	2986	2510	1.2	*ANXA5*	3396	3520	1.0
*CD44*	2947	1695	1.7	*COL1A1*	4127	3117	1.3
*CD49E (ITGA5)*	1064	1383	1.3	*MSX1*	3016	511	5.9
*CD73 (NT5E)*	1420	2207	−1.6	*PDGFRA*	266	2782	−10.5
*CD90 (THY1)*	630	1009	−1.6	*PDGFRB*	813	2739	−3.4
*CD105 (ENG)*	266	1824	−6.9	*PPARG*	196	240	−1.2
*CD106 (VCAM1)*	b	1748	−19.4	*RUNX2*	139	131	1.0
				*VIM*	11725	11777	1.0

b—below detection level.

**Table 2 ijms-20-04295-t002:** The mean microarray expression levels of differentially expressed genes classified as ECM-related according to the Panther algorithm.

Gene Symbol	iMPC	MSC	Fold Difference	Gene Name
*LRRC32*	131	993	−7.6	Leucine-rich repeat-containing protein
*GAS6*	364	2283	−6.3	Growth arrest-specific protein 6
*ADAMTS1*	294	1838	−6.3	A disintegrin and MMP with thrombosponding motifs 1
*VASN*	460	2516	−5.5	Vasorin
*ISLR*	b	417	−5.1	Ig superfamily containg leucine-rich protein
*SEPP1*	b	415	−4.8	Selenoprotein P
*EGFLAM*	b	365	−4.4	Pikachurin
*COL8A1 ^1^*	929	3854	−4.1	Collagen type VIIIα1
*CLEC3B ^1^*	b	336	−3.9	Tetranectin
*LTBP2 ^1^*	425	1402	−3.3	Latent TGFβ-binding protein 2
*CLEC11A1 ^1^*	158	495	−3.1	C-type lectin dmoain family 11 member A
*EFEMP2 ^1^*	876	2657	−3.0	EGF-containing fibulin-like ECM protein 2
*SCUBE3*	b	226	−2.6	Signal peptide CUB and EGF-like DCP3
*LAMB2*	317	794	−2.5	Laminin subunit beta-2
*ADAMTS5*	b	199	−2.4	A disintegrin and MMP with thrombospondin motifs 5
*LAMA3*	b	196	−2.3	Laminin subunit alpha 3
*AGRN*	284	126	2.3	Agrin
*FBLN2 ^1^*	904	177	5.1	Fibulin-2

^1^ Genes belonging to the subcategory ‘ECM structural proteins’; b—below detection level.

**Table 3 ijms-20-04295-t003:** Pathway-related Panther gene enrichment test based on all significantly differentially expressed genes and their fold expression difference.

Pathway	Number of Genes	*p*-Value
Integrin signaling pathway	24	0.0015
Opioid proenkephalin pathway	9	0.0023
Inflammation mediated by chemokine and cytokine signaling	21	0.0149
Plasminogen activating cascade	2	0.0158
Endothelin signaling pathway	7	0.0202
De novo purine biosynthesis	5	0.0209
Pyridoxal phosphate salvage pathway	2	0.0482

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
