# Peer review of "The Role of Extracellular Matrix Expression, ERK1/2 Signaling and Cell Cohesiveness for Cartilage Yield from iPSCs"

_ijms, 2019, doi:10.3390/ijms20174295_

Round 1

Reviewer 1 Report

The manuscript by Buchert et al., focused on the study of the geno-phenotype of MSC cells compared to iMPC cells and on their chondrogenesis and cartilage yield. The authors performed a transcriptome analysis, followed by bioinformatics analysis to identify ECM-related genes that were expressed at lower levels in iMPC cells and this could explain the reduced capacity of these cells to aggregate on the 2D surface. Additionally, iMPC cells showed lower activation of the pro-survival signaling ERK1/2.

The work is well conducted and presented. The data are of high quality. The results, discussion, and methods sections are well-reported and described. The data supported the conclusions of the manuscript.

I have some minor concerns:

1)    In the method section is mentioned that western blot analysis of pSmad2, Smad2/3, pSmad, Smad1, Smad5 were performed but the data are not presented;

2)    May the authors have some data on the transcriptome profile of MSC and iMPC cells implanted in vivo? Since the in vitro culture could influence the geno-phenotype of cells; this could be a limitation of the study that should be stated in the discussion.

Reviewer 2 Report

The manuscript „The role of extracellular matrix expression, ERK1/2 signaling and cell cohesiveness for cartilage yield from iPSCs“ from Justyna Buchert et al. characterizes very well the difference between MSCs and iMPCs with regard to chondrogenic differentiation. A mRNA array was performed – data are systematically analyzed and well presented. Selected experiments have been performed to prove hypotheses that arose from the array data. The manuscript is well organized and written. Only few remarks and questions remain:

It would be great if in figure 1E also MSC pellets could be shown for comparison. Please provide the ID of the Ethical vote including approval date. Line 429: β-mercaptoethanol Line 430: Germany is not necessary to mention again Line 434: How long was the incubation with the ROCK inhibitor? Please include a section on the statistical analysis performed. For comparison of two groups non-parametric test should have been used (usually Mann-Whitney U-test or Wilcoxon matched pairs test). Please provide primer information in a table – including gene ID, annealing temperature and PCR efficiency. If possible add a graphic of characteristic melting curve as supplementary information. Is it possible to name the positive controls used in supplementary table 3 and 4 (rule out that PCR failed)? Figure 3: Please show histograms for all 4 targets. And change figure legend which refers to light green lines – but greyscale picture is shown. Line 238: Unfortunately only preliminary data – would have been perfect to be included in the manuscript. Please double-check spacing between numbers and units. As limitation of the study it should be discussed that in this study iMPC derived from one iPS cell line was compared to primary MSCs from different donor. Bigger differences are expected between different primary cell donors than from repetitive experiments with one cell line.
